# The “Magic Bullet” Is Here? Cell-Based Immunotherapies for Hematological Malignancies in the Twilight of the Chemotherapy Era

**DOI:** 10.3390/cells10061511

**Published:** 2021-06-15

**Authors:** Nina Miazek-Zapala, Aleksander Slusarczyk, Aleksandra Kusowska, Piotr Zapala, Matylda Kubacz, Magdalena Winiarska, Malgorzata Bobrowicz

**Affiliations:** 1Department of Immunology, Medical University of Warsaw, 02-097 Warsaw, Poland; nina.miazek@wum.edu.pl (N.M.-Z.); slusarczyk.aleksander@gmail.com (A.S.); ola.kusowska@gmail.com (A.K.); matyldakubacz@gmail.com (M.K.); magdalena.winiarska@wum.edu.pl (M.W.); 2Institute of Physiology and Pathophysiology of Hearing, World Hearing Center, 05-830 Nadarzyn, Poland; 3Department of General, Oncological and Functional Urology, Medical University of Warsaw, 02-005 Warsaw, Poland; piotr.zapala@wum.edu.pl

**Keywords:** immunotherapy, chimeric receptor, NK cells

## Abstract

Despite the introduction of a plethora of different anti-neoplastic approaches including standard chemotherapy, molecularly targeted small-molecule inhibitors, monoclonal antibodies, and finally hematopoietic stem cell transplantation (HSCT), there is still a need for novel therapeutic options with the potential to cure hematological malignancies. Although nowadays HSCT already offers a curative effect, its implementation is largely limited by the age and frailty of the patient. Moreover, its efficacy in combating the malignancy with graft-versus-tumor effect frequently coexists with undesirable graft-versus-host disease (GvHD). Therefore, it seems that cell-based adoptive immunotherapies may constitute optimal strategies to be successfully incorporated into the standard therapeutic protocols. Thus, modern cell-based immunotherapy may finally represent the long-awaited “magic bullet” against cancer. However, enhancing the safety and efficacy of this treatment regimen still presents many challenges. In this review, we summarize the up-to-date state of the art concerning the use of CAR-T cells and NK-cell-based immunotherapies in hemato-oncology, identify possible obstacles, and delineate further perspectives.

## 1. Introduction

For many years, chemotherapy was the first-line treatment for blood cancers; however, such an approach is rarely curative and frequently leads to recurrence of the disease [1]. The very first form of immunotherapy, allogeneic hematopoietic stem cell transplantation (HSCT), followed by the development of monoclonal antibody (mAb)-based treatments, turned out to be particularly effective against hematological malignancies and encouraged further interest in immune cell-based therapies [2]. Although chemotherapy, HSCT, and mAbs remain the gold standard for blood cancer treatment, the development of novel, more effective immunotherapies is now at the forefront of ongoing research [3].

### 1.1. Standard Therapeutic Options in Hematological Malignancies

Cell-based immunotherapies provide an alternative to HSCT for patients with hematological malignancies with relapsed/refractory (r/r) disease. Nowadays, the main beneficiaries of CAR-based immunotherapy are r/r acute lymphoblastic leukemia (ALL) patients. Although over 80% of adult ALL patients obtain complete remission with intensive induction, consolidation, and maintenance chemotherapy, less than half of them achieve long-term leukemia-free survival, and the majority ultimately relapse [4]. The salvage therapy for such patients includes the use of immunotherapeutic approaches (e.g., mAbs targeting CD19/CD3 (blinatumomab) or CD22 (inotuzumab ozogamicin), CAR-T cell therapy) [5].

The management of diffuse large B-cell lymphoma (DLBCL) and follicular lymphoma (FL), the most common non-Hodgkin’s lymphomas (NHLs), relies on the use of rituximab-based immunochemotherapy (the so-called R-CHOP (rituximab, cyclophosphamide, doxorubicin, vincristine, and prednisolone)) that is successful in approximately 2/3 of the patients [6,7]. It has to be kept in mind that the r/r lymphoma patients generally have an extremely poor prognosis [8,9], as demonstrated by the SCHOLAR-1 trial, in which the objective response rate was 26% (complete response rate 7%) to the next line of therapy, and the median overall survival was 6.3 months for r/r DLBCL patients [10]. The studies involving whole-exome sequencing on large cohorts of primary DLBCL patients have defined the mutational landscape of lymphoma cells that may be responsible for r/r disease phenotype [11]. The recurrently mutated genes are linked to the regulation of apoptosis and cell proliferation, including BCL2 anti-apoptotic family members, MYC, and p53. However, so far there is limited evidence for beneficial effects of targeting these pathways by single targeting agents, suggesting the need for combination regimens [11].

In multiple myeloma (MM), which accounts for 10% of blood cancers, an increased understanding of disease biology has led to the emergence of three novel therapeutics, i.e., proteasome inhibitors, immunomodulatory drugs, and mAbs [12]. These new agents are currently the cornerstone of MM therapy and have almost completely replaced melphalan, an alkylating agent used for over 30 years as a standard in MM patients [13]. The novel agents are currently widely used in the treatment of MM, often in a combination involving two novel agents and a corticosteroid. However, myeloma remains largely incurable and patients with triple-class refractory disease (i.e., refractory to therapeutics from all three new groups) have limited treatment options and a dismal prognosis [13].

Another hematological malignancy that can potentially benefit from the introduction of cell-based approaches is chronic lymphocytic leukemia (CLL)-the most common adult leukemia, which, despite significant progress in its management, remains an incurable disease. While a large group of the patients will never require therapy, the majority of the treated patients experience a complete response (CR) or partial response (PR) to therapy followed by a relapse. The introduction of novel treatment modalities is especially important for patients with high-risk disease characterized by del17 (p13.1), p53 mutation, complex karyotype, or unmutated immunoglobulin variable regions which are associated with significantly shorter survival [14,15,16]. For many years, aggressive chemo-immunotherapy involving fludarabine, cyclophosphamide, and anti-CD20 rituximab (RTX) or obinutuzumab has been the preferred option [17]. Nowadays, the choice of drugs has been extended by kinase and BCL2 inhibitors that show high efficacy mostly irrespectively of the genetic background of the disease [18]. However, despite high response rates, only 10–30% of patients treated with these new agents achieve CR, even fewer reach negative minimal residual disease (MRD) status, and up to 50% eventually relapse within 3–4 years, resulting in limited options for further treatment and shortened survival [18,19,20].

As demonstrated above, the management of ALL, CLL, and NHL still mainly relies on the use of chemotherapy. Moreover, HSCT, which is a salvage therapy for a group of refractory patients, requires pre-treatment with aggressive chemotherapy as well. This places survivors at risk for the development of a diverse array of side-effects with the potential to negatively impact both their quality of life and ultimately, their survival [reviewed in [21,22]]. The most common of these long-term consequences are the development of a second malignancy [23,24] and cardiovascular complications [25,26,27]. Therefore, the development of modern, personalized, and more effective immunotherapies is currently being widely explored. 

### 1.2. Mechanisms of Immune Evasion

Since the concept of immunosurveillance was first proposed in the 1950s by Burnet [28,29], significant progress in understanding the role of the immune system and complex network of cellular interactions in the treatment of hematological malignancies has been made [30]. It is now evident that not only cytotoxic T cells but also other immune cell populations can participate in the elimination of tumor cells. Nowadays, the greatest challenge in cancer treatment is to overcome multiple evasion mechanisms allowing malignant cells to escape immune system recognition and hampering efficient tumor eradication [31]. Defects in antigen processing and presentation frequently arise as a result of a loss-of-function mutation in β2-microglobulin and downregulation of major histocompatibility complex (MHC) class I molecules, which prevent T cell activation and cytotoxic activity [32]. Moreover, decreased expression of MHC class II molecules impairs antigen recognition by CD4+ T cells and facilitates immune tolerance. Ineffective T cell activation in hematological malignancies often occurs as a result of upregulated cytotoxic T lymphocyte antigen 4 (CTLA-4) receptor engaging CD80/CD86 on antigen-presenting cells (APCs), leading to T cell anergy [33]. Another well-described antigen-escape mechanism is the overexpression of programmed death receptor ligand 1 (PD-L1) on leukemic cells interacting with PD-1 receptor on T cells, leading to inhibition of anti-tumor immunity and promoting tumor cell survival [34].

Furthermore, expansion of immunosuppressive cells modulates the inhibitory environment and suppresses the effector functions of T cells, contributing to tumor evasion in hematological malignancies. Accumulation of T-regulatory (Treg) cells facilitates immune suppression via direct cell-to-cell contacts (e.g., expression of PD-L1) and by secretion of immunosuppressive cytokines, i.e., interleukin (IL)-10, IL-35, transforming growth factor (TGF)-β, and production of granzymes and perforin, which lead to effector T cell apoptosis [35,36]. Similar effects are mediated by tumor-associated macrophages (TAMs) presenting anti-inflammatory M2 phenotype and overexpressing CD163, a marker correlating with poor prognosis in leukemia and lymphoma patients [37,38]. On the other hand, myeloid-derived suppressor cells (MDSCs), a heterogeneous population of immature myeloid cells, promote the formation of an immunosuppressive tumor environment by secreting anti-inflammatory cytokines and chemokines, nitric oxide (NO), and reactive oxygen species (ROS), thus hampering T cell infiltration and proliferation at the tumor site [39].

The mechanisms of immune evasion, presented above and depicted in Figure 1, which in consequence lead to the ineffective killing of the tumor cells by the main players in the tumor immunosurveillance-cytotoxic T and NK cells, may be successfully combated by the use of modern immunotherapy: 1) immune checkpoint targeting (elegantly reviewed in [40]), which is not the topic of this review and cell-based approaches comprising 2) chimeric antigen receptors (CAR) T cells and 3) NK cell-based immunotherapies, which will be further presented in this review paper.

## 2. CAR-T in Onco-Hematology

Adoptive immunotherapy has been introduced as a form of in vitro autologous or allogeneic immune cell stimulation with their subsequent administration directly back to the patient. One of the most advanced and personalized strategies of adoptive immunotherapy utilizes CAR-T cells, i.e., T cells genetically pre-modified to recognize tumor-associated antigens [41]. Conventional CAR-T treatment protocols consist of several consecutive steps. Firstly, autologous T lymphocytes are isolated from the patient’s blood by leukapheresis, then modified ex vivo with CAR-encoding viral vectors, multiplied, and finally reinfused directly into the patient. Most protocols require a single infusion, whereas time from collection to administration usually does not exceed 3 weeks [42,43,44]. 

To achieve full activation, physiological T cells require two signals. Signal 1 is initiated by TCR-CD3 binding to antigen presented in conjunction with MHC molecules on APCs. To prevent anergy, TCR signaling is modulated and amplificated by interactions of co-stimulatory domains (mainly CD28, 4-1BB, and OX40), delivering signal 2. Both, signal 1 and signal 2, trigger intracellular signaling pathways that result in activation, proliferation, cytotoxicity, and persistence of T cells [45]. Importantly, CAR constructs recognize antigens irrespectively of MHC presentation. Due to built-in co-stimulatory domains that mimic co-stimulation by ligandspresent on APCs, the first and second signal are initiated simultaneously after antigen recognition. Signal transduction via CAR leads to the recruitment of kinases and activation of modified cells [46,47,48].

After binding the tumor-associated antigen, CAR-T cell is fully activated through the intracellular activation and co-stimulatory domains (CD3ζ and typically 4-1BB or CD28) [49]. Because the following process provides both signal 1 and signal 2, T cell activation can be fully achieved [50,51].

To date, four generations of structurally different CAR-T cells have been developed. The first generation of receptors consists of the extracellular direct antigen-recognition domain and intracellular activation domain (CD3ζ) [50]. Insufficient stability and activation resulting from lack of co-stimulation is the main limitation of the first-generation CAR-T cells [52]. To enhance persistence and cytotoxicity, the second-generation CARs are equipped with additional co-stimulatory domains (most frequently CD28 or 4-1BB (CD137)) in tandem with the CD3ζ chain. Due to the transduction of signal 1 and signal 2 via CARs, the modified cells are characterized by increased proliferation, cytokine secretion (e.g., IL-2), and expression of anti-apoptotic proteins [50,53]. Based on the second-generation CAR-T cells, further genetic modifications to increase anti-cancer efficacy and optimize clinical effects were introduced. The third-generation CARs consist of a combination of co-stimulatory domains (CD3ζ-CD28-OX40 or CD3ζ-CD28-4-1BB, and DAP10). The third-generation CAR-T cells are primarily capable of enhanced proliferation, high cytokine production, increased tumor cell elimination, and enhanced expression of anti-apoptotic Bcl-xl protein. However, their functions do not simply combine the properties of the two additional domains and may largely depend on the dominant one [54,55,56]. The latest, still in the initial research phase, the fourth-generation CARs, also called TRUCKs (T cells redirected for antigen-unrestricted cytokine-initiated killing), can interact with the tumor environment by inducible or constitutively expressed interleukins (IL-2, IL-7, IL-12, IL-15, IL-18, IL-23) or ligands (CD40-L). TRUCKs are being studied especially in the treatment of solid tumors with diverse phenotypes of cancer cells [57]. Although the vitality of re-infused lymphocytes decreases with time, they can be still detected in patients’ blood from several weeks to several years following the infusion. It has been proposed that they are likely to survive even longer in the bone marrow or lymph nodes, but these tissues are not routinely analyzed [58,59,60].

Clinical studies have revealed that CAR-T cell expansion and persistence can be improved by prior lymphodepletion with different chemotherapy regimens, fludarabine and cyclophosphamide being the most commonly used [61]. The underlying mechanism is unclear; however, increased levels of IL-15 and an associated decrease in regulatory lymphocyte activity seem to play a crucial role in the persistence of reinfused cells [58,62,63,64].

### 2.1. Registered CAR-T Approaches

Promising outcomes in the clinical use of CAR-T therapy in hemato-oncology have already resulted in five products approved by the U.S. Food and Drug Administration (FDA) to date. The current registered CAR-T cell formulations involve four CD19-targeting products used in the treatment of B-cell-derived acute leukemias and aggressive lymphomas, and one compound targeting B-cell maturation antigen (BCMA) used in MM treatment (Table 1).

Tisagenleucel was the first registered CAR-T product [72] based on the results from the ELIANA study. Eighty-one percent of B-cell ALL patients who failed two lines of therapy and were subsequently treated with tisagenleucel achieved a 3-months overall response (OR), whereas 60% achieved CR [42]. Based on the encouraging results of the multicenter, phase 2 ZUMA-1 study, the FDA approved axicabtagene ciloleucel (axi-cel) for the second-line treatment of DLBCL. In multicenter phase 2 trials, axi-cel achieved 52–82% objective response rate and 40–54% of CR rate in DLBCL, mediastinal B-cell lymphoma, and FL [43], which resulted in the extension of FDA approval of tisagenleucel for DLBCL as the third-line therapy [42,73]. It should be mentioned, however, that the study group was advanced in the disease, which is supported by the fact that 30% of patients did not receive CAR-T infusion at all, due to disease progression and death [42]. Nevertheless, these two registrations have opened the entry of CAR-T products into the therapy of B-cell lymphomas. Recently, two novel products have been registered—brexucabtagene autoleucel for mantle cell lymphoma (MCL) and lisocabtagene maraleucel for DLBCL [74,75].

Results of the first clinical trials suggested that CD19-directed CAR-T cells may bring benefits for CLL patients; however, the percentage of complete responses is significantly lower than in DLBCL or B-ALL patients [76,77]. Other studies indicate that prior lymphodepletion may be crucial in obtaining a therapeutic response [78]. Encouraging outcomes were achieved by complementing treatment with tisagenlecleucel together with ibrutinib in a pilot clinical trial involving CLL patients [79]. Data from another clinical trial in CLL suggested that ibrutinib can modulate phenotype and ex-vivo autologous T cell expansion [80], but the mechanism of its positive effect on CAR-T therapy is yet to be thoroughly investigated.

Clinical safety and efficiency of BCMA-specific CAR coupled with CD3ζ and 4-1BB signaling domains were investigated in a phase 1 study (NCT02546167) on 25 patients with advanced refractory MM [81]. BCMA-targeting construct-JNJ 4528 achieved impressive OR and CR, reaching almost 100% and 69%, respectively, in the CARTITUDE-1 trial, [82] prompting the FDA to grant breakthrough therapy designation for MM with at least three prior lines of therapy [83,84]. The extremely promising results of both studies suggest that in the future BCMA-CAR-T therapy might play a crucial role in MM therapeutic protocols. However, further studies on long-term efficacy and comparison of CAR-based therapy with other currently available treatments are required.

### 2.2. Experimental CAR-T Approaches Targeting Other Antigens

The success of CD19-targeting agents prompted further efforts to introduce this immunotherapy in the treatment of other hematological malignancies. Although the applicability of CD19-directed CAR-T therapy in acute myeloid leukemia (AML) is limited, alternative leukemic cell-specific antigens such as CD33, CD38, CD56, CD117, CD123, Lewis-Y, and Muc-1 are currently being explored [85].

CD38, a transmembrane glycoprotein upregulated on MM cells, represents an attractive target for immunotherapies and has encouraged the development of anti-CD38 mAbs, i.e., daratumumab, isatuximab, and MOR202 [86]. Despite the great success of mAbs, management of relapsed or refractory MM patients still represents the major challenge due to the poor prognosis and decreased survival after anti-CD38 mAb treatment failure [87]. Therefore, CD38-CAR-T cell therapy has been proposed as an alternative cell-based approach. A profound cytotoxic effect of CD38-CAR on MM cell lines (RPMI8226, KMM1) was demonstrated in vitro, whereas cytotoxicity against patient-derived MM cells exceeded 90% [88]. Similar results were observed in B-cell non-Hodgkin lymphoma (B-NHL) cell lines, in primary cells from lymphoma patients, and in lymphoma NOD/SCID mice models, in which CD38-CAR efficiently eradicated tumor cells [89]. Nevertheless, the issue of the protective effect exerted by bone marrow mesenchymal stem cells (BMMSC) on the lytic activity of CD38-CAR in MM remains a challenge and reflects the need to generate high-affinity CAR constructs equipped with CD28/4-1BB co-stimulatory domains to preserve efficient CD38-CAR cytotoxic activity [90]. Another approach, treatment with a bispecific CD38-CAR coupled with anti-BCMA single-chain variable fragment, resulted in an overall response in 14 (87.5%) r/r MM patients with manageable toxicity in phase 1 clinical trial [91]. Other clinical trials involving CD38-CAR (NCT03464916) and BCMA/CD138/CD38/CD56-targeted CAR-T cells (NCT03473496) in r/r MM are still ongoing. Moreover, another emerging immunotherapy for MM is based on the generation of invariant natural killer T (iNKT) cells modified with CD38- or BCMA-CAR construct. Both CAR-transduced iNKT cells demonstrated potent cytotoxic activity against MM cell lines in vitro and efficiently lysed patient-derived CD38-positive MM cells [92].

Despite variable expression, CD38 has been considered as a potential therapeutic target also in AML. In vitro study with CD38-CAR-NK cells showed remarkable eradication of CD38-positive AML cell lines (THP-1, U937) and enhanced cytotoxicity against primary AML blasts. Moreover, improved anti-tumor activity was observed particularly upon pre-treatment of AML cells with all-trans retinoic acid, which promoted upregulation of CD38 on malignant cells and hence increased sensitivity toward CD38-CAR-NK treatment [93].

CD22 is another target explored in the treatment of B-cell lymphomas and leukemias in several ongoing clinical studies (i.e., NCT03999697, NCT04546906, NCT04088890). CD22 is widely expressed on the majority of pre-B ALL cells. In initial reports, CD22 CAR-T cells have been introduced as a potent and safe drug for patients with pre-B cell ALL, including individuals resistant to anti-CD19 immunotherapy [94].

Since the potential efficacy of CD20-targeting CARs is supported by the history of success with monoclonal anti-CD20 antibodies [95,96], CD20 has also been postulated as a therapeutic target for CAR-T therapy in NHL. The first proof-of-concept clinical trial of the first-generation CD20-CAR was performed in 2008 on seven patients with relapsed or refractory indolent B-cell lymphoma or MCL [97]. Although the safety and tolerability of this approach were demonstrated, the ex vivo expansion methods and persistence of the modified T cells in the organism were modest due to the lack of co-stimulatory domains [97]. Therefore, the same group produced the third-generation CD20-CAR containing CD28 and 4-1BB signaling domains and demonstrated promising results in terms of tolerability in a small (4 patients) pilot clinical trial (NCT00621452) [97]. The indispensability of the co-stimulatory domain was demonstrated by several preclinical studies using established B-ALL and NHL cell lines as well as in murine models [98,99,100] that further supported the use of CD20 as a promising target for CARs. Furthermore, CD20-CAR equipped with CD28 co-stimulatory domain and CD3ζ moiety was shown to effectively lyse several lymphoma and leukemia cell lines with downregulated CD20 expression as well as rituximab- or ofatumumab-refractory primary CLL cells [101]. Because the first-line treatment of CD20-positive B-cell malignancies often involves administration of RTX, the efficacy of CD20-CAR in a model of refractory lymphoma failing CD20-targeted therapy has been investigated. The results demonstrated unimpaired proliferation and preserved cytolytic activity of CD20-CAR-T cells in the presence of RTX against NHL cell lines in vitro. However, cytokine secretion and T cell cytotoxicity gradually declined with an increase in RTX concentration. Anti-tumor activity of CD20-CAR was retained in vivo in NSG lymphoma mice treated with rituximab, resulting in substantial tumor regression (52 days survival) in contrast to mice treated with RTX alone, which died by day 24. This indicates that residual RTX does not affect CD20-CAR efficacy and a low CD20 level is sufficient to mount CD20-CAR-based immune response [102]. Another construct where CD20-CAR domain was coupled with 4-1BB and CD3ζ moiety demonstrated efficacy in a small (7 patients) clinical trial (NCT01735604) in refractory advanced DLBCL patients [103]. The feasibility and efficacy of CD20-CAR treatment for NHL patients were further supported by the results of phase 2 of the latter study [104].

In response to antigen downregulation or loss manifested by CD19-negative relapses common in B-ALL and much less reported in lymphoma patients [105,106], the generation of multispecific CAR-T cells simultaneously targeting several antigens has become a prospective therapeutic strategy [107]. Recently, trispecific CD19/20/22-CAR-T cells that target B-ALL cells irrespectively of CD19 loss both in vitro and in an NSG xenograft model have been proposed as a salvage therapy or even front-line therapy in B-ALL [108]. The efficacy of such an approach has been confirmed in another study using murine models injected with a mixture of triple-positive cells, CD19-negative, CD20-negative, and CD22-negative B-cell lymphoma cell lines [109]. In this study, only the trispecific CAR-T cells rapidly and efficiently eliminated the tumors, while each of the monospecific CAR-T cells failed to prevent tumor progression [109]. Similar studies were conducted using CD19/CD20 bispecific CAR-T cells in in vitro models using primary CLL and B-ALL cells [110]. Prevention of antigen escape was also reported in NSG mice injected with Burkitt lymphoma Raji cells upon treatment with CD19/CD20-CAR, in contrast to mice treated with CD19-CAR, in which tumor outgrowth occurred [111]. In phase 1 clinical trial (NCT03019055), administration of bispecifc CD19/20-CAR-T cells resulted in CR among 64% and PR in 18% of NHL and ALL patients, simultaneously showing low toxicity [112]. CD19/CD22-CAR-T-cell therapy demonstrated promising efficacy in phase 1 clinical trial (NCT03185494) in adult patients with ALL or DLBCL [113].

Recently, CD37, a transmembrane protein highly expressed on malignant B cells, has received particular attention as a novel, alternative target for immunotherapies [114]. CD37-CAR demonstrated profound anti-tumor activity in a range of B-cell lymphoma cell lines and in patient-derived MCL xenograft (PDX) models in NSG mice [115]. The results from this study prompted the launch of a currently ongoing phase 1 clinical trial (NCT04136275) to assess the safety and efficacy of CD37-CAR in B-cell malignancies for the first time in humans. Effective inhibition of tumor progression was also observed in CD19-negative lymphoma cells treated with another CD37-CAR construct, in contrast to CD19-CAR, which failed to eradicate malignant cells [116]. As a way to overcome the issue of CD19 loss, a bispecific CD37/CD19-CAR has been proposed, and showed potent cytotoxicity against CD37-positive lymphoma cells in vitro, decreased tumor burden, and improved survival rate in a Raji-xenograft tumor NSG mice model [117].

Recently, the results of an interesting attempt to create a “universal CAR” were published [118]. It has been postulated that the generation of a CAR directed against CD126 (IL-6 receptor), an antigen that is broadly present on the surface of many hematologic and solid tumors (including MM, lymphoma, AML, pancreatic and prostate adenocarcinoma, non-small cell lung cancer, malignant melanoma, and many others), may be a solution to the extremely expensive production of CAR-T cells, which is partly due to the restricted use of each CAR construct for specific tumors. Thus, the development of a universal agent might facilitate a massive production process. CD126-directed CAR demonstrated cytotoxicity against a plethora of human malignant cell lines as well as MM and prostate cancer in murine models. Of note, CD126 is expressed at much higher levels on the surface of normal than malignant tissues. CD126-CAR had no cytotoxic effect against normal immune cells including B-, T-, and NK cells. The possible limitation of this approach relies on the relatively high expression of CD126 on the hepatocytes. In fact, transaminitis and possible hepatotoxicity remain an adverse effect that sometimes compromises the use of tocilizumab—an IL-6R-directed monoclonal antibody. However, to date, the results from murine models have not reported hepatotoxicity of CD126 CAR-T cells [118].

### 2.3. Limitations

#### 2.3.1. Toxicities

A significant limitation of the use of CAR-T cells is their high toxic potential. The most common side effects include cytokine release syndrome (CRS) and immune effector cell-associated neurotoxicity syndrome (ICANS). CRS is triggered by the recognition of cognate antigen followed by the activation of T cells and bystander immune cells (mostly macrophages), which results in a massive release of a wide range of inflammatory cytokines such as IL-1, IL-6, IL-15, and interferon (IFN)-γ [119,120]. These excessively secreted pro-inflammatory cytokines induce a generalized immune activation that can be manifested by symptoms classified by a 4-grade scale, from mild symptoms such as headache, nausea, myalgia, fatigue, or fever occurring at 1 grade to 4-grade life-threatening multi-organ dysfunction [121,122,123]. Similarly to tumor lysis syndrome occurring after chemotherapy, severe CRS more commonly affects patients with bulky disease and correlates with the efficacy of CAR therapy [51,124].

Neurologic manifestations occur almost exclusively in patients developing CRS, mostly at the same time but may also appear prior to CRS. More aggressive clinical manifestation of CRS is usually associated with higher neurological toxicities [125,126]. Although the pathogenesis of neurotoxicity is poorly understood, the most likely trigger of this syndrome is the extensive diffusion of cytokines through the blood-brain barrier and trafficking of T cells into the central nervous system [127,128]. ICANS manifestations include mild symptoms such as cognitive defects, language disturbance and impaired handwriting, tremors, delirium, seizures, disturbance of consciousness, or dysphasia but also could be lethal [122,123,126,129]. Fortunately, in most non-lethal cases, the symptoms are completely reversible with no residual defects [123].

The crucial complication associated with the treatment of both syndromes is a limited arsenal of symptomatic drugs. Simultaneous use of tocilizumab mitigates CRS without inhibiting CAR-T cell activity [43,123,130,131]. Therefore, treatment with glucocorticosteroids or tocilizumab should always be carefully considered and, if possible, previous serum levels of IL-2, IL-6, IL-10, IFN-γ, and C-reactive protein (CRP) should be assessed [121,122,123,132]. Despite poor penetration into the cerebrospinal fluid, tocilizumab can also be used in stage 2 of the symptoms’ severity and higher, although its effectiveness is lower. In the case of stage 3 and 4 symptoms, intensive care unit admission and steroid administration are required [51,123,133]. It is worth mentioning that the severity of CRS incidence corresponds to the effectiveness of CAR-T therapy [51,124]. Other, less common side effects include B-cell aplasia (short or long-term) with concomitant hypogammaglobulinemia, graft-versus-host disease (GvHD) in patients with stem-cell transplantation prior to CAR-T treatment, and "off-tumor" cytotoxicity toward targets non-specific for malignant cells, only [134]. Although the loss of B cells can be managed with intravenous immunoglobulin infusions [135], targeting more abundant molecules, e.g., human epidermal growth factor receptor 2 (HER2), may lead to serious on-target off-tumor cytotoxicity as observed in a clinical trial of HER2-directed CAR where severe potentially fatal side effects were reported [136].

#### 2.3.2. Strategies to Overcome CAR-T Limitations

As already mentioned, one of the limitations of the use of CAR-T cells is their potentially life-threatening side effects. Thus, there is a need to modify CAR constructs in order to introduce “safe switches” that would enable switching off the effector cells. The recent advances in pharmacological control of CAR-T cells have been elegantly reviewed by Caulier et al. [137].

Attempts to mitigate CAR-T cell adverse effects have been made via co-modification of CAR-T cells with surface antigens that could be targeted with therapeutics, such as rituximab and cetuximab [138,139]. However, as antibody-mediated cytotoxicity may be too slow to efficiently eliminate CAR-T cells in case of fulminant side-effects, modification of CAR-T cells with inducible caspase 9 (iCasp9) may be a solution enabling rapid removal of these cells. It is a potent tool that leads to the elimination of >90% of engineered T cells within 30 min [140]. In this system, the iCasp9 gene contains the intracellular portion of the human caspase 9 protein, a pro-apoptotic molecule, fused to a drug-binding domain derived from human FK506-binding protein. The activation of apoptosis is executed upon intravenous administration of a small-molecule drug AP1903, resulting in cross-linking of the drug-binding domains, caspase-9 dimerization, and activation of apoptosis executioner, caspase 3 [141]. For example, incorporation of the iCasp9 system in CAR construct targeting SLAMF7, a protein highly expressed on MM cells, enabled rapid AP1903-induced CAR-T cell elimination in vitro and substantially reduced CAR-T cell number in solid MM tumor NSG mice upon AP1903 treatment, as compared to control mice lacking the “suicide switch” [142]. The utility of the iCasp9 system is currently being investigated in several clinical trials recruiting blood cancer patients, who will receive allogeneic stem cell transplants. In interventional clinical trials (NCT01494103, NCT01744223, NCT00710892) insertion of the “suicide” iCasp9 gene into donor T cells aims to introduce sensitivity to AP1903 and to enable effective elimination of T cells causing GvHD, however preserving them in sufficient numbers for cancer eradication. Similarly, phase 1/2 dose-escalation study in r/r CD19-positive B-lymphoma patients is about to evaluate the safety and efficacy of the iCasp9 system in CAR-based treatment involving CD19-CD28-CD3 ζ-iCasp9-IL15-transduced cord blood natural killer (CB-NK) cells (NCT03056339). However, this strategy leads to an irreversible halt to therapy, which is not desirable in cases of progressive disease [143]. Therefore, “suicide” gene activation is suggested to be used as the last resort. Hence, the use of a tyrosine kinase inhibitor—dasatinib, that suppresses the activation of TCR signaling kinases and impairs T cell cytotoxicity—may be an option to temporally halt cytotoxicity of CAR-T cells [144].

Another strategy to overcome on-target off-tumor toxicity relies on attributing specificity to antigens expressed not only on target tumor cells, but also on healthy counterparts. This approach has been tested in preclinical studies in mice and rhesus macaques [145]. Because CD33 molecule is expressed on healthy and neoplastic myeloid cells, it is not an optimal target for CAR therapy and may result in off-tumor toxicity and destruction of healthy myeloid cells. Kim et al. [145] have provided evidence that CD33 depletion from normal hematopoietic stem and progenitor cells (HSPC) prior to autologous HSPC transplantation can generate a functional hematopoietic system that allows specific targeting of AML using CD33-directed CAR-T cells.

Potential solutions to overcome the antigen specificity issue relies on redirecting CAR cytotoxicity towards labeled antibodies or small fluorescein-based adapters that recognize various tumor-associated epitopes [146]. In such an approach, CAR-T cells are generated to recognize a tag or fluorescent dye, and therefore their cytotoxic effect relies on antibody-dependent cell-mediated cytotoxicity (ADCC) mechanism [147,148]. This allows not only the generation of universal CARs to use in various malignancies, but also may provide a tool to overcome intra- and intertumoral heterogeneity [149].

The issue of limited controllability and flexibility of conventional CAR-T therapies has been recently addressed by the development of adapter CAR-T cells, a system combining T cells engineered with adapter CAR and soluble adapter molecules, where extracellular tumor-targeting and signaling domains are uncoupled (review in [150]). The greatest advantage of this strategy is the possibility to precisely control the activity of adapter CAR-T cells by modulating the level of adapter molecules within the body. An in vitro study on an NK-92 cell line modified with adapter CAR utilizing biotinylated monoclonal antibodies (b-mAb) as adapter molecules demonstrated significant cellular cytotoxicity against a range of lymphomas—NHL and MCL cell lines—as well as patient-derived MCL and CLL cells [151]. A similarly designed adapter CAR-T system based on an scFv fragment capable of recognizing endogenous vitamin biotin coupled with adapter, i.e., b-mAb, led to potent T cell activation and malignant cell lysis upon recognition of single antigen or multiple antigens by simultaneously utilizing several adapters [152].

#### 2.3.3. Financial Aspects

Extremely complex preclinical preparation translates into the very high cost of CAR-T treatment. Depending on the pre-treatment protocol and severity of side effects, the total cost of CAR-T therapy ranges from USD 30,000 to 900,000, constituting the main barrier to broad use of CAR-T [153,154,155]. The main logistic problem arises from the required access to highly specialized laboratories. To reduce the time from bench to bedside, CAR-T cells from allogeneic universal donors are currently being investigated [156,157]. Standardization and optimization of collection and production of CAR-T therapy are believed to reduce time to reinfusion and increase the effectiveness of treatment.

Currently, the efforts of researchers are focused on improving the therapeutic capacity of CAR-T therapy by, for example, pre-modifying lymphocytes to drive the secretion of pro-inflammatory cytokines, e.g., IL-12. Another approach includes combining CAR-T with antibodies blocking PD-1 and/or CTLA4. The current challenges in improving the safety and efficacy of the modern CAR constructs are elegantly reviewed in [146,158] and [143].

## 3. NK Cell-Based Immunotherapies

As tumor cells evade the immune response by downregulation of self-antigens and MHC class I molecules, which otherwise could be recognized by T cells, one of the natural weapons to be used to eliminate tumor masses are natural killer (NK) cells. NK cells constitute a subset of innate cytotoxic lymphocytes with an ability to recognize and kill virally infected or tumor-transformed cells without prior activation [159]. The role of NK cells in cancer immune surveillance was underlined in multiple studies demonstrating NK cell cytotoxic activity towards cancer cells [160]. Moreover, pro-inflammatory cytokines (IFN-γ, TNF-α) and chemokines (CCL3, CCL4, and CCL5) secreted by NK cells stimulate other lymphocyte populations in the tumor microenvironment or exert a direct anti-tumor effect [160].

NK cell cytotoxicity might be elicited in two distinct mechanisms—direct, which relies on the balance between stimulatory and inhibitory signals obtained from encountered cells [160], and indirect, which requires prior target cells opsonization with antibodies (ADCC). In this mechanism, NK cells recognize an Fc antibody fragment through the CD16 receptor (FcγRIIIa) and consequently release cytolytic granules and cytokines, leading to cancer cell death. In both mechanisms, immunological synapse formation facilitates the trafficking of secreted granzymes and perforin into a target cell or death receptor stimulation (e.g., FAS, TRAIL) leading to target cell apoptosis [159]. However, unfortunately, the function of NK cells frequently becomes compromised by the tumor immune escape mechanisms (reviewed in [161,162]).

The lower numbers of NK cells in peripheral blood have been shown as a negative prognostic factor in DLBCL [163]. Other studies demonstrated that the anti-leukemic activity of NK cells inversely correlated with disease progression in AML, where NK cell suppression at leukemia diagnosis and relapse were observed more frequently [164]. It also has been shown that in AML, overexpressed aryl hydrocarbon receptor (AHR) transcription factor induces miR-29b expression in NK cells, thereby impairing NK cell maturation and function [165]. NK cell function impairment was also explored in Hodgkin lymphoma (HL), where it correlated with elevated serum levels of soluble ligands for NK cell receptors NKp30 (BAG6/BAT3) and NKG2D (MICA), which are known to constrict NK cell function [166].

### 3.1. NK Cell Harnessing Strategies

Different approaches have been implied to implement NK cells into therapeutic regimens for cancer. Autologous NK cell transfer is associated with limited clinical efficacy despite successful NK cell persistence in peripheral blood [167]. Conversely, allogeneic NK cell infusions exert higher efficacy due to KIR-ligand mismatch. It has been demonstrated that AML patients treated with haploidentical related donor NK cells received complete hematological remission [168]. NK cell haploidentical transfer performed as a bridge therapy to HSCT in patients with AML and high-risk myelodysplastic syndrome (MDS) led to complete remissions (in 6 out of 16 patients) [169]. Haploidentical infusion of NK cells, which were primed with the CTV-1 leukemia cell line lysate, demonstrated efficacy in AML patients, leading to complete remission in some (3/12 cases) [170].

Enhancement of NK cell in vivo activity remains a challenge and has been addressed in multiple clinical trials including IL-2 and IL-15 administration. The main disadvantage of IL-15 administration was frequent adverse effects (fever, thrombocytopenia, hypotension), whereas IL-2 infusion benefits were hampered by concurrent T regulatory cell expansion [171,172]. Recombinant IL-15 constitutes a promising alternative with more potent and selective activity. Of note, a murine study demonstrated the high efficacy of IL-15 superagonist complex ALT-803 in enhancing NK cell function and development of effector NK cells and CD8+ T-cell responders of the innate phenotype [172]. ALT-803 was also shown to significantly increase NK and CD8+ T-cell numbers and function in relapsed allo-HSCT patients [173]. Administration of ALT-803 in patients with advanced solid tumors (melanoma, kidney, head and neck, lung cancer) led to NK cell expansion and was characterized by an acceptable safety profile [174].

It is worth noting that some of the other already registered cancer drugs might be also utilized to restore NK cell function. Such a phenomenon has been observed for lenalidomide, bortezomib, GSK3 inhibitors, and tyrosine kinase inhibitors (e.g., imatinib, sunitinib, sorafenib) [175,176]. For instance, lenalidomide lowers the NK cell activation threshold and enhances IFN-γ production [177].

### 3.2. Checkpoint Blockade and ADCC Enhancement Strategy

Various antibody-based therapies have been introduced in order to augment NK cell anti-tumor activity. One of the strategies encompasses NK cell inhibitory receptor blocking. Anti-KIR2D mAb-lirilumab administered to patients with different malignancies (phase 1 trial) exhibited an acceptable safety profile and full KIR occupancy [162]. Patients with relapsed or refractory HL and MM receiving nivolumab plus lirilumab reached a 76% OR rate. Treatment responses were also observed in MM patients treated with IPH2101, another anti-KIR mAb, combined with lenalidomide [161]. IPH2101 also revealed potential utility in AML treatment in elderly patients [162]. Notably, in murine studies, KIR blockade was confirmed to enhance the efficacy of anti-CD20 antibodies, currently used as a therapeutic standard [178]. Another monoclonal antibody, anti-NKG2A monalizumab, already has been evaluated in solid tumors (advanced gynecological malignancies), but not yet in leukemia [179]. In preclinical evaluation, immunodeficient mice with leukemia infused with NKG2A(+) NK cells were pre-treated with anti-human NKG2A and thus rescued from disease progression [180]. The ability of monalizumab to restore direct cytotoxicity of NK cells against HLA-E-expressing CLL targets provides another justification for its possible clinical implementation [181]. Of note, PD-1/PD-L1 axis blockade might also augment NK cell anti-tumor responses, which already have been shown in vivo [182]. A significant contribution of NK cells to the anti-tumor activity of anti-CTLA-4 antibodies in combination with oncolytic virus infection was also observed [183].

The role of NK cells in the clinical effect of mAbs has been widely described in attempts to develop antibodies with the increased capability of NK cell activation [184]. Antibodies targeting tumor antigens (anti-BAFF-R, anti-CD123, anti-CD157, anti-SLAMF7, anti-GD2, anti-CD33) with an enhanced affinity toward CD16, which further increases ADCC efficacy, were evaluated in multiple clinical trials [185].

Other strategies to unleash NK cell anti-cancer effects are still being evaluated in preclinical models. They involve antibody-mediated MICA and MICB shedding inhibition (7C6 mAb) and other antibodies (e.g., anti-CD133, anti-IL-7) with enhanced ability to induce ADCC [186,187,188].

### 3.3. Off-the-Shelf NK Cell Transfer

Although NK cells do not need priming to target and kill tumor cells, they typically are isolated in low numbers and tend to be short-lived. Hence, NK cell donation and ex vivo expansion from cancer patients are troublesome. Therefore, the idea of NK-92 human cell line administration as a widely available source of allogeneic NK cells has been proposed. NK-92 infusion in r/r AML patients was characterized by an acceptable safety profile but limited response [189]. On the other hand, some encouraging responses were observed in lung cancer patients [190]. NK-92 cells also can be further engineered with CD16 for increased ADCC or with CAR. Accordingly, NantKwest in several preclinical settings and clinical trials is currently testing PD-L1 t-haNK cells expressing CAR targeting PD-L1 [191,192]. Studies demonstrated the ability of PD-L1 t-haNK to eliminate human cancer cell lines of different origin, including triple-negative breast cancer (TNBC) and lung, urogenital, and gastric cancer cells, as well as both monocytic and granulocytic MDSCs within the tumor microenvironment [193,194,195]. Moreover, another option for off-the-shelf therapy is the collection of NK cells from healthy donors, followed by their expansion and modification with mbIL-15, which helps to maintain the proliferation of NK cells for a long time. This approach is used by Nkarta Therapeutics to manufacture allogeneic NK cells targeting either NKG2D ligands expressed on a variety of both hematologic and solid malignancies or CD19 antigen to treat B-cell malignancies. Off-the-shelf NK cell cancer immunotherapy is also being actively explored by Fate Therapeutics, where allogeneic NK cells are derived from a clonal master iPSC line and further engineered with high-affinity CD16, IL-15 receptor fusion, or CAR (CD19, BCMA, B7H3, MICA/B).

### 3.4. Specific Killer Engagers

Novel antibody constructs, namely, bispecific killer engagers (BiKEs) and trispecific killer engagers (TriKEs), are promising agents facilitating selective anti-tumor NK cell cytotoxicity. The above-mentioned agents include two or three scFvs linked together, respectively. BiKEs and TriKEs designed to engage tumor antigens (e.g., CD19, CD20 for B cell NHL; CD33 and CD123 for AML; CD30 for HL), and CD16 receptor facilitate NK cell immunological synapse formation. One of the scFvs might be replaced by cytokines (e.g., IL-15) exhibiting NK cell stimulatory properties [196]. The first clinical trial evaluating anti-CD16 x IL-15 x anti-CD33 TriKE safety in hematological malignancies is ongoing (NCT03214666).

### 3.5. CAR-NK Concept Outsmarts CAR-T Cells

CAR-T cell therapy has shown remarkable clinical efficacy albeit accompanied by severe adverse effects. Exploiting CAR-NK cells might be an alternative with a higher safety profile and comparable efficacy. The procedure of effector cell preparation and administration to the patient is similar in the case of both CAR-T and CAR-NK cells (Figure 2).

CAR-NK therapy is associated with a lower risk of autoimmune reactions and fatal CRS [171]. To date, CAR-NK cells have been administered to r/r CD19-positive cancer patients (CLL or NHL) and demonstrated a promising response rate (73%) with no major toxicity [197]. Another study demonstrated that CD33-CAR-NK-92 cell infusions can be safely applied with no substantial adverse effects in patients with r/r AML [198]. Other trials evaluated the efficacy of CAR-NK in B-ALL, but the results are still unpublished (NCT01974479, NCT01974479). Anti-CD22 and anti-CD19/CD22 CAR-NK trials are planned (NCT03692767, NCT03824964). In preclinical studies, different approaches were proposed to enhance CAR-NK efficacy and safety. One of them is simultaneous transduction with IL-15 and “suicide switch” iCasp9. IL-15 improves CAR-NK function, whereas iCasp9 provides safety control [197]. Another preclinical study demonstrated that B-ALL can be effectively targeted by FLT3-specific CAR-NK cells and rapidly inactivated by iCasp9 induction [199].

## 4. Perspectives

CAR-based therapy seems to be one of the most promising and future-oriented immunotherapies. Nevertheless, there is still room for improvement and a better understanding of the anti-tumor mechanisms of CAR-modified effector cells. Nowadays, CAR-based approaches can fill the treatment gap and benefit mainly patients with relapsed/refractory lymphoma or leukemia incurable with standard available protocols. However, complete remissions in those patients appeared to be followed by a relatively high relapse rate related to antigen loss and lineage switch. Although many steps were undertaken to prevent these relapses, it became clear that more comprehensive knowledge is crucial to understand the biology of both tumor cells and CAR cells to achieve a satisfactory compromise between treatment efficacy and the possibility of cytotoxic effects. Therefore, several approaches are currently being developed that can influence CAR-T cell fitness and persistence in patients including the adjustment of CD4/CD8 ratio, T cell phenotype, and CAR domains. Moreover, a recently emerging approach to overcome the selection of antigen-negative subclones is the combinatorial targeting by, for example, dual-specificity CARs.

Another issue associated with CAR therapy seems to be the lack of tumor-specific antigens and the necessity to target tumor-associated antigens that are also expressed on healthy tissues. The vast majority of antigens utilized in immunotherapy for leukemia or lymphoma are B cell-specific proteins, expressed by both malignant and normal B cells but rarely present in other tissues. The resulting on-target off-tumor toxicity to normal B cells is unavoidable but manageable with intravenous immunoglobulin infusion. However, in the case of solid tumors, there is a strong need for developing additional strategies sparing healthy tissues, such as logic gates, adjusting the threshold for CAR-T cell activation, using inhibitory or inducible CARs, or incorporating “off -switches” that deactivate or eliminate CAR-T cells attacking healthy cells.

The non-negligible disadvantage of CAR-T therapy is the cost and time needed for the preparation of CAR-T products from autologous T cells. Therefore, the use of allogeneic CAR-T cells from donors or CAR-NK cells has become a promising alternative to standard CAR-T therapy. Notable advantages of such approaches are the immediate off-the-shelf availability and possible standardization of products that, together with decreased costs, would significantly increase access to this class of therapeutics.

Despite all the above-mentioned challenges, CAR-based approaches have already revolutionized the field of immunotherapy for hematological malignancies and remain one of the most promising approaches in the treatment of cancer [200].

## Figures and Tables

**Figure 1 cells-10-01511-f001:**
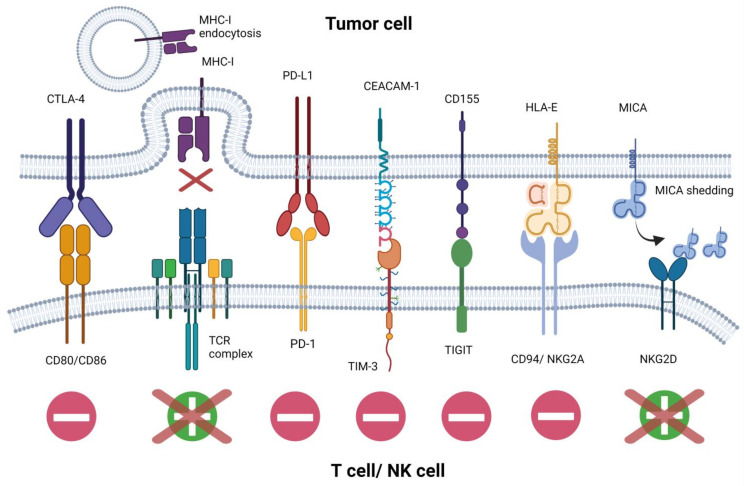
Mechanisms of tumor cell evasion from NK and T cell surveillance. Tumor cells can employ a plethora of immunosuppressive membrane antigens to constrict NK cell and T cell function. Antigens eliciting tumor-specific immune response undergo endocytosis or shedding, whereas inhibitory immune checkpoint molecules expressed on tumor cells hamper NK cell and T cell function.

**Figure 2 cells-10-01511-f002:**
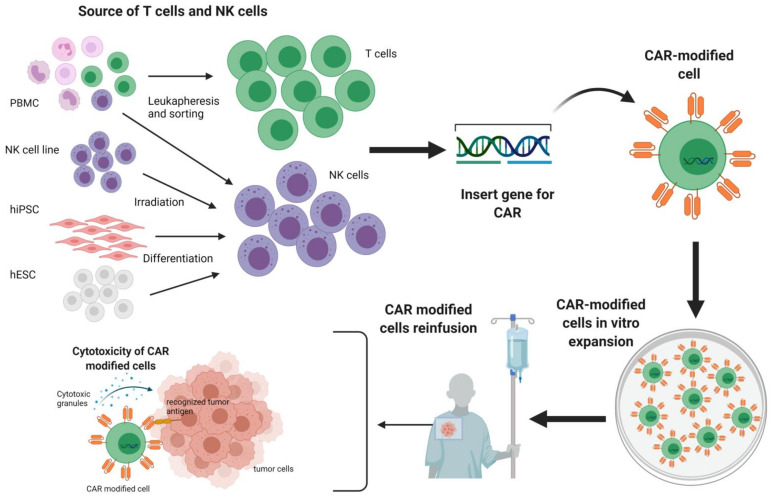
The outline of therapy with CAR-modified T cells and NK cells. Both T cells and NK cells can be obtained from the peripheral blood by leukapheresis; alternatively, NK cells also can be derived from human induced pluripotent stem cells (hiPSC), human embryonic stem cells (hESC), and irradiated NK cell lines. Subsequently, T cells or NK cells are modified with a gene for chimeric antigen receptor (CAR) and expanded in vitro for many days. Finally, CAR-modified cells are re-infused into the patient and exert specific anti-tumor cytotoxicity.

**Table 1 cells-10-01511-t001:** FDA approved CAR-T products.

Name	Molecular Target	Intracellular Activation Domain	Indication	Registration Date	Registration Basis
tisagenlecleucel (Kymriah)	CD19	41BB-CD3ζ	B-ALL, DLBCL	Aug 2017	ELIANA trial in B-ALL (NCT02435849) [42,65] JULIET trial in DLBCL (NCT02445248) [66]
axicabtagene ciloleucel (Yescarta)	CD19	CD28-CD3ζ	DLBCL, FL	Oct 2017	ZUMA-1 trial (NCT02348216) [67,68]
brexucabtagene autoleucel (Tecartus)	CD19	CD28-CD3ζ	MCL	Jul 2020	ZUMA-2 trial (NCT02601313) [69]
lisocabtagene maraleucel (Breyanzi)	CD19	41BB-CD3ζ, also contains a nonfunctional truncated epidermals growth factor receptor (EGFRt)	DLBCL	May 2021	TRANSCEND trial (NCT02631044) [70]
idecabtagene vicleucel (Abecma)	BCMA	41BB-CD3ζ	MM	Mar 2021	NCT02658929, NCT02546167 [71]

B-ALL: acute B lymphoblastic leukemia, DLBCL: diffuse large B-cell lymphoma, FL: follicular lymphoma, MCL: mantle cell lymphoma, MM: multiple myeloma, BCMA: B-cell maturation antigen.

## Data Availability

Not applicable.

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
