# Peer review of "The “Magic Bullet” Is Here? Cell-Based Immunotherapies for Hematological Malignancies in the Twilight of the Chemotherapy Era"

_cells, 2021, doi:10.3390/cells10061511_

Round 1

Reviewer 1 Report

This review article summarized the recent updates on the CAR cell therapy in hematological malignancies. This whole article was nicely written and the information provided was up-to-date and detailed. I believe this article will be useful for the field. However, there are still some minor comments might make the article better:

  1. Line 93, “Since the following process provides both signal 1 and signal 2”. Please first introduce signal 1 and signal 2 here. For example, what is the signaling cascade and how they lead to full T cell activation?
  2. Line 95-104, please introduce in more details about the differences on each generation. For example, how is cytotoxicity or cytokine production enhanced? What is the mechanism?
  3. Line 432, “3.3. Off-the-shelf NK cell transfer”, please extend this section more with details.

Author Response

We would like to thank the Reviewer for time and consideration. We agree that addressing the issues pointed out by the Reviewer improves the quality of our review. Specifically, we have introduced extended information on signal 1 and 2 and T cell activation. Moreover, we introduced a thorough description of differences between generations of CAR constructs. Finally, the section 3.3 entitled Off-the-shelf NK cell transfer has been extended. We believe that these changes will make the manuscript more attractive to the readers.

Reviewer 2 Report

In the review entitled “The “magic bullet” is here? Cell-based immunotherapies for hematological malignancies in the twilight of the chemotherapy era”, the authors comprehensively summarize current CAR T-cell and NK-cell immunotherapeutic approaches in hematologic malignancies.

The following revisions may be beneficial for improving the review.

  • The current role of chemotherapy and other standard therapies for treatment of hematological malignancies (benefits and limitations) could be outlined more intensively to better meet the intension of the manuscript title “Cell-based immunotherapies for hematological malignancies in the twilight of the chemotherapy era”.
  • Authors could also provide a separate section on current solutions / approaches to overcome the limitations of CAR-T cell therapy e.g. suicide genes, adaptor CARs etc.
  • The manuscript should be checked for consistency . E.g. usage of ORR, switch between BE and AE (tumour vs. tumor).

Author Response

We would like to thank the Reviewer for the important comments that could help us to improve the quality of our work. We have addressed all the issues raised by the Reviewer. Thanks to the Reviewer’s suggestion we have introduced a brief introduction on the currently used standards of therapy in hematological malignancies that are potential candidate for cell-based therapies. Moreover, the current approaches to overcome the limitations of CAR-T cell therapy have been summarized. Finally, the manuscript has be spell-checked and all inconsistencies between British and American English have been standardized to AE.